# Resistance of Lung Cancer to EGFR-Specific Kinase Inhibitors: Activation of Bypass Pathways and Endogenous Mutators

**DOI:** 10.3390/cancers15205009

**Published:** 2023-10-16

**Authors:** Ilaria Marrocco, Yosef Yarden

**Affiliations:** 1Department of Life Sciences and Public Health, Università Cattolica del Sacro Cuore, 00168 Rome, Italy; ilaria.marrocco@unicatt.it; 2Department of Immunology and Regenerative Biology, Weizmann Institute of Science, Rehovot 76100, Israel

**Keywords:** adaptive mutagenesis, EGFR, kinase inhibitor, lung cancer, mutation, receptor tyrosine kinase, resistance to drugs

## Abstract

**Simple Summary:**

Genome-based cancer medicine is becoming the standard of care: the patient’s tumor DNA is first analyzed to identify driver mutations, and this permits later selection of the most effective drugs. Treatment of lung cancer offers many examples. Activating mutations in the epidermal growth factor receptor (EGFR) gene, as well as in other receptor tyrosine kinases (e.g., ALK), are considered actionable candidates, and the respective drugs, called tyrosine kinase inhibitors, are relatively effective. Unfortunately, despite initial activity, the emergence of new, on-target mutations, along with adaptive processes, preempt the anti-cancer effects and necessitate switching to next-generation drugs. This review highlights recent progress in resolving the mechanisms that underlie acquisition of resistance. Specifically, we focus on the endogenous mutators that initiate emergence of new mutations and the potential clinical benefits that may be derived from this new understanding.

**Abstract:**

Epidermal growth factor receptor (EGFR)-specific tyrosine kinase inhibitors (TKIs) have changed the landscape of lung cancer therapy. For patients who are treated with the new TKIs, the current median survival exceeds 3 years, substantially better than the average 20 month survival rate only a decade ago. Unfortunately, despite initial efficacy, nearly all treated patients evolve drug resistance due to the emergence of either new mutations or rewired signaling pathways that engage other receptor tyrosine kinases (RTKs), such as MET, HER3 and AXL. Apparently, the emergence of mutations is preceded by a phase of epigenetic alterations that finely regulate the cell cycle, bias a mesenchymal phenotype and activate antioxidants. Concomitantly, cells that evade TKI-induced apoptosis (i.e., drug-tolerant persister cells) activate an intrinsic mutagenic program reminiscent of the SOS system deployed when bacteria are exposed to antibiotics. This mammalian system imbalances the purine-to-pyrimidine ratio, inhibits DNA repair and boosts expression of mutation-prone DNA polymerases. Thus, the net outcome of the SOS response is a greater probability to evolve new mutations. Deeper understanding of the persister-to-resister transformation, along with the development of next-generation TKIs, EGFR-specific proteolysis targeting chimeras (PROTACs), as well as bispecific antibodies, will permit delaying the onset of relapses and prolonging survival of patients with EGFR^+^ lung cancer.

## 1. Introduction

Along with the well-characterized roles for growth factors and their receptor tyrosine kinases (RTKs) in development, physiology and regeneration, these cell-surface molecules have important roles to play in tumorigenesis. Two major modes underlie the ability of growth factors and RTKs to initiate or promote cancer progression. The first involves genetic aberrations of specific RTKs. For example, 10–30% of patients with lung cancer present activating mutations in the epidermal growth factor receptor (EGFR) gene [1,2,3,4,5]. Importantly, the presence of EGFR mutations predicts response to specific tyrosine kinase inhibitors (TKIs), likely due to addiction of the respective tumor cells to an active EGFR pathway [6]. The most frequent gene aberration of HER2/ERBB2, the closest homologue of EGFR, is gene amplification, which occurs in breast, gastric and other tumors, but rare missense mutations have also been reported [7,8]. Importantly, very high expression levels of HER2 predict response to chemotherapy combined with HER2-targeting monoclonal antibodies (mAbs), such as trastuzumab/Herceptin. Similarly, overexpression of EGFR occurs in approximately 50% of brain tumors of glial origin due to gene amplification [9], and a large fraction of these tumors also present internal deletions within the extracellular domain [10]. However, these observations have not been translated to EGFR-targeted treatments of brain malignancies. The other oncogenic roles of growth factors and RTKs involve autocrine or paracrine loops that engage a growth factor and a specific RTK. For example, high expression of EGFR ligands, especially amphiregulin, supports proliferation of colorectal cancer (CRC) cells and might predict response to a combination of chemotherapy plus an anti-EGFR mAb, such as cetuximab [11]. However, the abundance of amphiregulin in CRC specimens has not been translated to a predictive biomarker of response to anti-EGFR antibodies. Yet another clinically approved combination of chemotherapy and an antibody blocking a ligand of RTKs is the combination of bevacizumab/Avastin and chemotherapy, which is used to treat patients with recurrent ovarian cancer. Similar combinations of bevacizumab were approved for breast cancer, non-small-cell lung cancer, glioblastoma, renal cell carcinoma and cervical cancer [12]. Notably, aside from its mitogenic activities, VEGF plays a major role in controlling blood vessel formation (angiogenesis) as well as modulating tumor-induced immunosuppression.

## 2. Major Classes of Oncogene-Targeted Drugs and Resistance to the Respective Therapeutic Strategies

Only two major classes of molecular targeted treatments currently dominate the field of RTK-centered cancer treatment. These are antibodies targeting either growth factors or RTKs [13], and small-molecule kinase inhibitors, which block tyrosine-specific and other kinases [14]. A third class of inhibitors, which can selectively sort specific proteins for intracellular degradation, is already on the horizon. The key for this strategy is the development of proteolysis-targeting chimeras (PROTACs). While some of the RTK-targeting drugs have greatly changed the landscape of the respective diseases, for example trastuzumab in breast cancer and osimertinib/Tagrisso in lung cancer, drug resistance is considered a major issue. Indeed, overcoming resistance to the latest anti-cancer drugs has been recognized by the Cancer Moonshot Initiative as one of the top ten priorities of contemporary cancer research [15]. This is because the majority of patients with advanced cancer die either because their cancers are inherently resistant to drugs, or their tumors initially respond but they later develop drug resistance [16]. Resistance universally limits applications of many different drugs, including not only chemotherapeutic agents but also kinase inhibitors [16,17], anti-receptor antibodies [13] and immune checkpoint blockers [18,19]. This review will focus on the emergence of resistance of lung cancer to anti-EGFR kinase inhibitors, highlighting the underlying mechanisms and potential strategies to prevent cancer relapses.

## 3. A Primer to Lung Cancer

Nearly 2 million deaths are expected annually due to lung cancer, the most common cause of cancer-related deaths. Histologically, lung cancer can be divided into two groups: (i) Small-cell lung cancer (SCLC) constitutes approximately 15% of all cases. This deadly tumor is highly different from other lung cancers [20]. Its aggressive nature is attributable to bi-allelic inactivation of TP53 and RB1, along with aberrantly active Notch signaling. (ii) Non-small cell lung cancer (NSCLC) represents the remaining 85% of lung cancers and includes three major histological subtypes: adenocarcinoma, the most common type, squamous cell carcinoma and large cell carcinoma, the least frequent subtype. Unlike squamous NSCLC, which rarely harbors actionable mutations, several such mutations are highly prevalent in adenocarcinomas and include mutant forms of the following oncogenes: *KRAS*, *EGFR*, *ALK*, *RET*, *ROS1*, *BRAF*, *HER2* and *MET* [21]. Although immunotherapy combinations, for example the combination of antibodies targeting CTLA-4 and PD1, seem to benefit some patients presenting with advanced or metastatic NSCLC, tumors with EGFR mutations in general lack abundant infiltrating lymphocytes and have a relatively low tumor mutational burden [22]. Hence these tumors tend not to respond to immune checkpoint blockers. This contrasts with most KRAS- and BRAF-mutated NSCLCs, which are associated with a higher mutational burden [23].

## 4. EGFR Mutations in Lung Cancer

EGFR and its family members, primarily HER2, are ubiquitously expressed in epithelia, including the epithelium of the lung, and they have been found to be overexpressed or mutated in several types of cancer [24]. Depending on age, gender and ethnicity, 10–30% of all patients with NSCLC present somatic kinase-activating mutations in the gene encoding EGFR [1,2,3,4,5]. Another factor that might increase incidence of EGFR-mutated lung cancer appears to be air pollution. Hill et al. showed that exposure to particulate matter measuring ≤2.5 μm (PM2.5) promotes lung cancer development by recruiting macrophages into the lungs, which in turn release IL-1β. Consequently, a progenitor-like cell state is induced in lung alveolar type II epithelial cells harboring pre-existing EGFR activating mutations [25].

Various short deletions in exon 19 (Del19), along with L858R, a point mutation in exon 21, represent 85–90% of all known *EGFR* mutations in lung cancer [26]. Rare mutations account for 10–15% of all *EGFR* mutations in NSCLC and include G719A or G719S, Del18, E709K, exon 19 insertions, S768I, L861Q and exon 20 insertions (Ins20). Two mutations, T790M and C797S, typically emerge when patients are treated with the first/second or third generation TKIs, respectively [27,28,29,30]. It is imperative to note that the mutant forms of EGFR mimic the canonical mechanisms permitting activation of the kinase domain in response to EGF binding. The mutant forms also engage the same signaling pathways, primarily JAK-to-STAT, the RAS-to-mitogen-activated protein kinases (MAPK) pathway and the linear route that enables activation of mTOR by the upstream kinases PI3K and AKT. Although normally kinase activation targets EGFR to CBL-mediated receptor ubiquitination and subsequent degradation in lysosomes, certain EGFR mutants escape this regulation and also display enhanced heterodimerization with HER2, which further leads to persistent stimulation [31]. In addition, while wild-type EGFRs require dimerization for proper signaling [32,33], all kinase-activating mutations, except for L858R, induce an active conformation of the enzyme that is independent of ligand-induced dimerization [34].

## 5. First and Second Generations of EGFR-Specific Kinase Inhibitors (See Table 1)

It is interesting to draw analogies between TKIs targeting EGFR and TKIs specific to BCR-ABL1, a fusion tyrosine kinase that acts as the driver of chronic myeloid leukemia (CML). The development of imatinib, a BCR-ABL1 inhibitor, allows patients with CML to experience near-normal life expectancy [35]. Although specific point mutations that decrease drug-binding affinity can produce imatinib resistance, second- and third-generation TKIs can largely mitigate this issue. For patients with advanced NSCLC who are treated with a third-generation EGFR inhibitor, the median survival is greater than 3 years [36], significantly better than the <2 year survival rate just a decade ago. When compared to standard chemotherapy, the reversible ATP-competitive first-generation EGFR inhibitors, erlotinib and gefitinib, prolonged progression-free survival (PFS) and overall survival of patients with NSCLC who harbor mutant forms of EGFR [37,38,39]. In comparison to chemotherapy, EGFR TKIs are relatively safe drugs. Nevertheless, several side effects have been reported. They frequently include skin effects and gastrointestinal tract toxicity, such as skin rash and diarrhea, respectively. More severe adverse effects include intestinal obstruction, hepatotoxicity and interstitial lung disease [40]. Unfortunately, despite initial efficacy, treated patients eventually evolve drug resistance. Several mechanisms of acquired resistance have been described, including the gatekeeper T790M mutation within the EGFR kinase domain [27]. Importantly, T790M occurs in >50% of patients who relapse after treatment with the first-generation inhibitors. This mutation increases the affinity of the mutant receptor for ATP, thus preventing drug binding and reducing the potency of ATP competitive drugs. Less frequent routes of evasion have been identified, including amplification of genes encoding compensatory RTKs, such as *MET* [41] and *HER2* [42], as well as overexpression of the hepatocyte growth factor (HGF) [43,44], or up-regulation of another RTK, AXL [45]. The second-generation EGFR inhibitors, such as afatinib and dacomitinib, both bind with EGFR in an irreversible manner, aimed at providing therapeutic answers to T790M-mediated resistance [46,47]. However, although two second-generation TKIs have been approved for clinical use, these drugs also inhibit HER2 and HER4, and they frequently fail to prevent the emergence of T790M in patients [48].

**Table 1 cancers-15-05009-t001:** EGFR-specific agents, including clinically approved drugs, targeting mutant forms of EGFR in NSCLC.

Drug	Inhibitor Type	Drug Target	Status	Relevant Studies
Gefitinib	1st generation TKI Competitive Reversible	Del19/L858R-EGFR	Approved by the FDA/EMA in 2003/2009	NEJ002 [49,50] IPASS [51,52] WJTOG3405 [37,53]
Erlotinib	1st generation TKI Competitive Reversible	Del19/L858R-EGFR	Approved by the FDA/EMA in 2004/2005	OPTIMAL [39,54] ENSURE [55] EUTARC [38]
Icotinib	1st generation TKI Competitive Reversible	Del19/L858R-EGFR	Approved in China in 2011	CONVINCE [56] EVIDENCE [57]
Afatinib	2nd generation TKI Covalent Irreversible	Del19/L858R-EGFR	Approved by the FDA/EMA in 2013	LUX-Lung3 [58,59] LUX-Lung6 [59,60]
Dacomitinib	2nd generation TKI Covalent Irreversible	Del19/L858R-EGFR	Approved by the FDA/EMA in 2018/2019	ARCHER1050 [61,62]
Osimertinib	3rd generation TKI Covalent Irreversible	Del19/L858R/T790M-EGFR	Approved by the FDA/EMA in 2015/2016	AURA3 [63] FLAURA [64]
Aumolertinib *	3rd generation TKI Covalent Irreversible	Del19/L858R/T790M-EGFR	Approved in China in 2020	AENEAS [36,65] APOLLO [66]
Furmonertinib ^$^	3rd generation TKI Covalent Irreversible	Del19/L858R/T790M-EGFR	Approved in China in 2021	FURLONG [67]
Lazertinib	3rd generation TKI Covalent Irreversible	Del19/L858R/T790M-EGFR	Approved in South Korea in 2021	LASER201 [68,69] LASER301 (NCT04248829)
Befotertinib	3rd generation TKI Covalent Irreversible	Del19/L858R/T790M-EGFR	Clinical, Phase II/III (active)	NCT03861156 [70] NCT04206072 [71]
Abivertinib ^£^	3rd generation TKI Covalent Irreversible	Del19/L858R/T790M-EGFR	Clinical, Phase I/II (active)	NCT02274337 [72] AEGIS-1 (NCT02330367)
Nazartinib	3rd generation TKI Covalent Irreversible	Del19/L858R/T790M-EGFR	Clinical, Phase I/II (active)	NCT02108964 [73]
Mavelertinib	3rd generation TKI Covalent Irreversible	Del19/L858R/T790M-EGFR	Clinical, Phase I/II (terminated)	NCT02349633 [74]
Rociletinib	3rd generation TKI Covalent Irreversible	Del19/L858R/T790M-EGFR	Rejected by the FDA in 2016	NCT01526928 TIGER-1 (NCT02186301) TIGER-3 (NCT02322281) [75]
Olmutinib	3rd generation TKI Covalent Irreversible	Del19/L858R/T790M-EGFR	Terminated °	NCT01588145 NCT02485652 [76]
Naquotinib	3rd generation TKI Covalent Irreversible	Del19/L858R/T790M-EGFR	Clinical, Phase III (terminated)	NCT02588261 [77]
EAI001	4th generation TKI Allosteric Reversible	L858R/T790M/C797S-EGFR	Preclinical	[78,79]
EAI045	4th generation TKI Allosteric Reversible	L858R/T790M/C797S-EGFR	Preclinical	[78,79]
JBJ-04-125-02	4th generation TKI Allosteric Reversible	L858R/T790M/C797S-EGFR	Preclinical	[80]
JBJ-09-063	4th generation TKI Allosteric Reversible	L858R/T790M/C797S-EGFR	Preclinical	[81]
CH7233163	4th generation TKI Non covalent Competitive	Del19/L858R/T790M/C797S-EGFR	Preclinical	[82]
BLU-945	4th generation TKI Reversible	Del19/L858R/T790M/C797S-EGFR	Phase I/II (Recruiting)	SYMPHONY (NCT04862780)
BBT-176	4th generation TKI Reversible	Del19/L858R/T790M/C797S-EGFR	Phase I/II (Recruiting)	NCT04820023 [83]
TQB3804	4th generation TKI	Del19/L858R/T790M/C797S-EGFR	Phase I (Unknown)	NCT04128085
BPI-361175	4th generation TKI	Del19/L858R/T790M/C797S-EGFR	Phase I/II (Recruiting)	NCT05329298
HJM-561	PROTAC	Del19/L858R/T790M/C797S-EGFR	Preclinical	[84]
DDC-01-163	PROTAC	L858R/T790M/C797S-EGFR	Preclinical	[85]
Mobocertinib	TKI	Ins20-EGFR	Approved by the FDA in 2021	EXCLAIM [86]
Amivantamab	Bispecific Antibody	Ins20-EGFR MET	Approved by the FDA/EMA in 2021	CHRYSALIS [87] PAPILLON (NCT04538664)

The abbreviations used are: TKI, tyrosine kinase inhibitor; Del19, deletions in exon 19; FDA, Food and Drug Administration of the United States of America; EMA, European Medicines Agency. *, previously known as almonertinib; ^$^, previously known as alflutinib; ^£^, also known as avitinib; °, approved in South Korea in 2016 and subsequently terminated following two cases of toxic epidermal necrolisis (one fatal); PROTAC, proteolysis targeting chimera; Ins20, insertions in exon 20.

## 6. Third-Generation TKIs (See Table 1)

The third-generation EGFR inhibitors were designed to specifically inhibit EGFR-T790M, while sparing wild-type *EGFR* [88,89]. Similarly to the second-generation drugs, the newer compounds irreversibly bind with the mutant receptor by covalently engaging cysteine 797. Of the three compounds that initially entered pre-clinical tests, WZ4002, rociletinib and osimertinib, only the latter has been approved, in 2015. Although rociletinib demonstrated clinical activity in patients who had relapsed after treatment with the first and second generation TKIs [90], it was eventually discarded due to adverse effects and low efficacy. Two phase III studies established osimertinib as the current drug of choice for patients who progressed under the first-generation TKIs, as well as the drug of choice in the first line scenario. The first trial [63] compared the efficacy of osimertinib and platinum plus pemetrexed-based therapy in patients who had disease progression after first-line EGFR-TKI therapy. In all patients, the median duration of progression-free survival was significantly longer with osimertinib than in the other arm and, likewise, the objective response rate was significantly better with osimertinib. Additionally, the median duration of progression-free survival was longer among patients who received osimertinib, and the proportion of patients with moderate or severe adverse events was lower with osimertinib. The other trial was similarly successful [64]. It compared osimertinib and standard EGFR-TKIs in patients with previously untreated EGFR mutation-positive advanced NSCLC. The patients received either osimertinib or a first-generation TKI. While both arms showed similar safety profiles, the efficacy of osimertinib was superior to that of the standard (first-generation) EGFR-TKIs in the first-line setting. These observations led to the approval of osimertinib not only for patients who progressed following treatment with first-generation TKIs but also as a first-line drug for patients harboring EGFR-activating mutations.

## 7. Resistance to Osimertinib and Development of Fourth-Generation EGFR Inhibitors (See Table 1)

Despite impressive therapeutic effects, emergence of resistance to osimertinib is nearly inevitable. The most common mechanism of resistance to second-line osimertinib comprises *MET* amplification (18%), followed by a tertiary mutation within the ATP-binding pocket of the receptor, namely C797S (14%) [91]. Other resistance mechanisms include amplification of HER2, as well as mutations in downstream molecules (i.e., PIK3CA, KRAS and BRAF) [91]. Additionally, approximately half of the patients who progressed following osimertinib treatment within the AURA 3 trial revealed loss of T790M, and at least a third of these patients displayed concurrent resistant mechanisms (e.g., MET, HER2 or PIK3CA amplification) [91]. Moreover, histological transformation from NSCLC to SCLC has been described as one of the mechanisms responsible for osimertinib resistance [92]. In a similar way, the FLAURA 3 trial that tested osimertinib in a first-line setting detected the C797S mutation but no T790M [93]. In addition, this trial identified several off-target mechanisms that might lead to resistance, including amplification of MET and KRAS, HER2 exon 20 insertion and mutations in MEK1, JAK2, KRAS and PIK3CA [93,94]. Note that C797S-EGFR was identified in circulating free DNA (cfDNA) prior to the availability of resistance biopsy specimens [30]. In comparison to the >50% of patients who evolve resistance to the first-generation TKIs due to T790M, the C797S mutation is typically detected in less than a quarter of patients who progress under second-line osimertinib [95]. Fourth-generation inhibitors have been developed with the goal of overcoming C797S-mediated secondary resistance. EAI045 is an allosteric, non-ATP competitive inhibitor able to inhibit not only C797S but also T790M [78]. This compound targets an allosteric site of EGFR but it requires the co-administration of an anti-EGFR antibody. Interestingly, EAI045 is active on L858R-expressing tumors, but not Del19-expressing tumors [78]. A similar compound, JBJ-04-125-02, can inhibit cell proliferation and the triple mutant L858R/T790M/C797S as a single agent [80]. Similarly, BLU-945, a fourth-generation ATP-competitive inhibitor, showed promising in vitro results against C797S-positive tumors [96]. An initial clinical trial is currently evaluating BLU-945 in the TKI resistance setting, as either monotherapy or combined with osimertinib (NCT04862780). Yet another promising fourth-generation TKI is UPR1444, a new sulfonyl fluoride derivative which potently and irreversibly inhibits the triple EGFR mutant through the formation of a sulfonamide bond with the catalytic residue Lys 745 [97]. In summary, several different fourth-generation inhibitors are already in clinical trials.

## 8. PROTACs

In parallel to the development of fourth-generation EGFR TKIs, a major current effort is directed towards clinical development of proteolysis-targeting chimeras (PROTACs), representing an alternative strategy to directly target EGFR. PROTACs consist of two covalently linked protein ligands: one binds with the target protein (the protein intended to be degraded), while the other molecular arm recruits an E3 ligase that promotes ubiquitination and degradation of the target protein [98]. Notably, the numbers of PROTACs that are currently being evaluated in vitro, in vivo and in clinical studies is increasing [99]. Several PROTACs targeting EGFR have been developed (reviewed in [100]). By using EGFR TKIs as the target protein binder, it is possible to selectively target mutated forms of the receptor, while sparing the wild-type form of EGFR. In this regard, two PROTACs synthetized by using gefitinib as the EGFR binder were able to reduce proliferation of Del19- and L858R-EGFR expressing NSCLC cells, along with the levels of EGFR and the activation of downstream signaling pathways [101]. In a more recent study, Vartak et al. developed a PROTAC containing cetuximab as the EGFR binder, thereby reducing viability of EGFR-mutated NSCLC cells [102]. Despite the promising results showing in vivo efficacy of EGFR-specific PROTACs [103], additional preclinical data are needed prior to translating these findings to clinical applications. For example, membrane permeability and metabolic stability are crucial factors that need to be considered before attempting clinical development of EGFR-specific PROTACs.

## 9. Darwinian Mechanisms Underlying Resistance to EGFR-Specific TKIs

In the context of anti-cancer therapies, Darwinian mechanisms of drug resistance refer to a selection process that relies on pre-existing genetic variation that is present in a cancer cell population before starting the treatment. The pressure applied by the drug allows cells expressing a resistance-conferring mutation to survive the treatment (selection), leading over time to the establishment of a drug-resistant cancer cell population. Darwinian selection was observed, for instance, in patients with KRAS wild-type CRC that developed resistance after treatment with the anti-EGFR monoclonal antibody panitumumab [104]. While KRAS mutations were undetectable prior to the treatment, they became detectable 5–6 months after treatment with the antibody.

To address the question how resistant clones evolve during TKI treatment, Hata et al. exposed NSCLC cells to a TKI and monitored the development of several resistant clones. This identified early emergence of clones, which represented pre-existing resistant T790M clones, as well as slowly emerging clones that represented de novo acquisition of the T790M mutation within initially T790-WT cells [105]. In a similar way, Turke et al. applied high-throughput FISH analyses on both cell lines and lung cancer patients, and identified subpopulations of cells with *MET* amplification prior to drug exposure [44]. It is worth noting that emergence of pre-existing mutations post treatment with a TKI has been well-characterized in imatinib-treated patients with Philadelphia chromosome positive acute lymphoblastic leukemia (Ph+ ALL). For example, baseline mutations were identified in 21% of imatinib-naïve patients with newly diagnosed Ph^+^ ALL [106]. In conclusion, the examples from lung cancer, CRC and leukemia exemplify the role played by pre-existing mutations and intra-tumor heterogeneity in resistance to anti-cancer drugs.

## 10. Non-Darwinian Mechanisms of Resistance and Drug Tolerance Persistence

Non-Darwinian selection includes all mechanisms that do not involve genomic changes. In this context, epigenetic alterations, factors from the microenvironments, cellular plasticity and phenotypic adaptation are phenomena that might lead to drug resistance. In analogy to antibiotics-tolerant sub-populations of bacteria [107], a small subpopulation of drug-tolerant persister (DTP) cells has been identified in vitro [108]. DTPs are characterized by slow proliferation, reversible phenotypic alterations and altered energy consumption ([109]; see Figure 1). The mechanisms that underlie persistence include diverse epigenetic and transcriptional programs. We recently studied the stepwise transition from DTPs to resisters and detected a host-dependent but non-mutational reversible resister state [110]. In the same vein, single-cell RNA sequencing identified in patient-derived xenografts a rare population of DTP-like cells that converted the major preexisting population of cancer-associated fibroblasts into a state that can promote DTP survival [111]. In analogy, resistance of CRC to cetuximab has been attributed to a transcriptional switch from a cetuximab-sensitive subtype to a fibroblast- and growth factor-rich subtype at progression [112]. These observations propose that DTPs can switch phenotypes or educate their microenvironment. Consistent with this, using a combination of large-scale drug screening and whole-exome sequencing, it was concluded that the DTP state might provide a latent reservoir of cells for the emergence of diverse drug-resistance mechanisms [113]. Although there is no consensus concerning markers of DTP phenotypes, they nevertheless seem to share common features. For example, resistance to various EGFR inhibitors might share aberrant activation of the RAS-to-ERK (extracellular signal-regulated kinase) signaling, and this might be caused by either chromosomal amplification of the *MAPK1* gene or by down-regulation of a phosphatase that down-regulates ERK [114].

Another common feature was unraveled by employing barcode labeling, which revealed that antioxidant profiles characterized by increased glutathione metabolism and decreased reactive oxygen species (ROS) are hallmarks of cycling persisters [115]. Similarly, it has been proposed that the epithelial to mesenchymal transition (EMT), a reversible program of trans-differentiation, generates the drug-tolerant cell populations characterized by activation of ABC transporters and AXL, as well as immune evasion and epigenetic reprogramming [116]. Yet another marker emerged from single-cell RNA-seq and single-cell ATAC-seq analyses which confirmed the previously characterized genes *AURKA*, *VIM* and *AXL*, but added *CD74*, a gene that undergoes up-regulation in the drug-tolerant state [117]. Yet another interesting marker of DTPs is the Wnt/β-catenin signaling pathway, which is activated in response to TKI treatment in a notch3-dependent manner, thereby leading to survival of a subpopulation of stem-like DTP cells [118]. The use of genetically engineered mouse models confirmed that the notch pathway can activate proliferation of TKI-treated cells, such that combined treatments using notch inhibitors plus TKIs could block tumor relapses [119]. In summary, the non-Darwinian mechanisms of drug resistance, especially the emerging common functional features of DTP cells, exemplify the robust and diversified nature of drug tolerance. This offers a spectrum of molecular targets, such as notch and AXL, for combination treatments able to nullify or significantly delay the onset of resistance to drugs, including EGFR inhibitors.

## 11. Endogenous Mutators Promote the Emergence of New Mutations While under TKI Treatment

In vitro studies uncovered two routes of resistance to TKIs: (i) a Darwinian route that selects specific clones, which are drug resistant due to a pre-existing secondary mutation, and (ii) a less understood route that promotes de novo emergence of resistance-conferring mutations [105]. Notably, the mechanisms underlying emergence of the secondary (e.g., T790M) and tertiary (e.g., C797S) EGFR mutations are distinct from the mode that generates primary EGFR mutations. The primary mutations likely pre-exist in normal lung tissues but they are later exposed due to the action of tumor promoters, such as air pollutants [25]. What mechanisms could underlie the apparent drug-induced accelerated evolution? In 1975, Radman reported an inducible bacterial DNA mutagenesis system, the SOS response (see Figure 2), which might explain the link between genotoxic stress and adaptive mutagenesis [120]. Following genotoxic stress, bacteria release fragments of single-stranded DNA, which act as sensors that initiate transcriptional programs and mutate the genome [121]. The major endogenous mechanism of mutagenesis (i.e., mutator) in *Escherichia coli* is DNA polymerase V (polV), which instigates virtually all SOS mutagenesis [122]. PolV belongs to the group called Y family DNA polymerases, which promote translesion synthesis (TLS) of DNA. These polymerases exhibit low fidelity, thereby increasing mutagenesis rates when they are engaged in DNA replication. We previously investigated whether the treatment of lung cancer with TKIs similarly engages hypermutators [123]. Because GAS6 (growth arrest-specific protein 6), AXL’s ligand, is up-regulated in cycling DTP cells and it binds with newly externalized phosphatidylserine of apoptotic bodies, we assumed that the GAS6-AXL module acts as an alarm that stimulates SOS-like reactions in response to TKIs. In line with this prediction and previous reports that associated AXL and GAS6 with intrinsic resistance to TKIs [110,124], we found that AXL overexpression can up-regulate low-fidelity DNA polymerases and down-regulate DNA repair enzymes [123]. Moreover, simultaneously inhibiting AXL and EGFR completely blocked relapses in animal models. Metabolomic analysis uncovered yet another intrinsic mutator that relates to the dependency of DNA replication on balanced pools of deoxyribonucleotides (dNTPs) [125]. By activating MYC and purine synthesis, AXL disbalances the pools of dNTPs, which can influence polymerase proofreading [126] and mutator phenotypes [127,128]. It is worthwhile noting that in similarity to NSCLC cells, human CRC cells exploit adaptive mutability to evade the therapeutic pressure of a combined treatment that used a BRAF kinase inhibitor and an anti-EGFR antibody [129]. This treatment down-regulated mismatch repair (MMR), as well as homologous recombination genes and concomitantly up-regulated error-prone DNA replication. In conclusion, pharmacological stress-induced mutagenesis (SIM) might be shared by eukaryotes and unicellular organisms, arguing against prevailing assumptions that mutations occur purely stochastically [130].

## 12. A Biomarker Predicting Response of EGFR^+^ Tumors to Antibody Rather Than TKI Treatment

Although genome-based, personalized cancer medicine is rapidly becoming the standard of medical oncology [131], and with the exception of exon 20 mutation carriers, all patients with EGFR+ NSCLC are treated in the same way. Contrary to this practice, meta-analyses of multiple randomized trials that compared TKIs and chemotherapy confirmed superiority of TKIs and, unexpectedly, revealed that the hazard ratio of progression-free survival for tumors with Del19 was 50% greater than the ratio calculated for tumors with L858R [132]. Importantly, most kinase-activating mutations induce an active conformation of the kinase domain that is independent of ligand-induced EGFR dimerization [133]. For example, exon 19 deletions, exon 20 insertions and the dual L858R/T790M EGFR mutant do not require receptor dimerization. This contrasts with the L858R mutant, which depends on dimerization [34]. These observations raised the possibility that L858R might serve as a biomarker able to predict responses of EGFR+ lung cancer to dimerization-blocking antibodies like cetuximab. As predicted, our recent analyses showed that cetuximab monotherapy completely inhibited relapses of L858R patient-derived xenograft models, but tumors harboring other mutations rapidly relapsed post treatment with either cetuximab or TKIs [134]. Interestingly, unlike TKIs, which elevated reactive oxygen species (ROS) and induced robust cell death, antibody treatments only modestly associated with apoptosis, but they accelerated the rate of EGFR degradation and down-regulated several RTKs that have previously been implicated in drug resistance [134]. Taken together, these observations warrant clinical tests aimed at different, mutation-based immunotherapeutic treatments that would limit the use of TKIs and avoid emergence of secondary EGFR mutations.

## 13. Individual Bypass Mechanisms and the Respective Combination Therapies (See Figure 3 and Table 2)

### 13.1. MET Activation

Resistance to the first-, second- and third-generation EGFR TKIs frequently involves activation of compensatory pathways due to amplification and/or overexpression of bypass survival receptors (e.g., HER2, HER3, MET, AXL and IGF1R). As a result, resistant tumors no longer depend on EGFR for survival and proliferation. The MET signaling pathway is the most commonly engaged pathway following treatment with EGFR inhibitors, regardless of TKI type and line of therapy [135]. It has been shown that amplification of the *MET* gene causes resistance via the HER3/PI3K/AKT pathway [41]. Several MET kinase inhibitors have been developed. Two inhibitors, capmatinib and tepotinib, have been approved for NSCLC carrying the MET exon 14 skipping mutation [136,137]. Additionally, antibodies targeting MET are currently being evaluated in both preclinical and clinical studies. Of note, a bispecific antibody co-targeting MET and EGFR, amivantamab, has been approved for the treatment of NSCLC patients expressing EGFR with exon 20 insertions [138]. Considering the relatively high frequency of MET alterations in lung cancers with EGFR mutations, co-targeting EGFR and MET with either a kinase inhibitor or an antibody appears to be a logical therapeutic strategy that is currently being evaluated. A phase Ib/II trial testing a combination of capmatinib and gefitinib in patients with EGFR-mutated NSCLC that developed resistance to EGFR TKIs showed encouraging results, particularly in tumors with high *MET* gene copy number [139]. Promising results were also obtained in a phase I trial that combined the MET small-molecule inhibitor savolitinib together with osimertinib and recruited patients with EGFR^+^ NSCLC harboring *MET* amplification post treatment with an EGFR inhibitor [140]. Yet another trial combined the EGFR/MET bispecific antibody, amivantamab, and a third-generation EGFR inhibitor, lazertinib [141]. The same drug combination is also being evaluated in a first-line therapy by a phase 3 study, MARIPOSA [142].

**Table 2 cancers-15-05009-t002:** Combinatorial approaches under clinical evaluation to overcome resistance to third-generation EGFR-TKIs.

Mechanism of Resistance	Strategy to Overcome Resistance	Drugs	Status	Relevant Studies
MET alterations	MET TKI	Osimertinib + savolitinib	Phase Ib (active)	NCT02143466
		Osimertinib + savolitinib	Phase II (recruiting)	NCT03778229 (SAVANNAH)
		Osimertinib + savolitinib	Phase III (recruiting)	NCT05015608 (SACHI)
		Osimertinib + savolitinib	Phase II (recruiting)	NCT03944772 (ORCHARD)
		Osimertinib + savolitinib	Phase II (not yet recruiting)	NCT05163249 (FLOWERS)
		Osimertinib + savolitinib	Phase III (recruiting)	NCT05261399 (SAFFRON)
		Osimertinib + savolitinib	Phase II (active)	NCT04606771
		Osimertinib + tepotinib	Phase II (active)	NCT03940703 (INSIGHT)
		Osimertinib + vebreltinib	Phase I/II (recruiting)	NCT04743505
	Bispecific Antibody (EGFR-MET)	Lazertinib + amivantamab	Phase III (active)	NCT04487080 (MARIPOSA)
		Lazertinib ± amivantamab	Phase I (recruiting)	NCT04077463 (CHRYSALIS2)
		Lazertinib + amivantamab	Phase III (recruiting)	NCT05388669 (PALOMA3)
		Lazertinib ± amivantamab ± carboplatin/pemetrexed	Phase I (recruiting)	NCT02609776 (CHRYSALIS)
		Lazertinib + amivantamab + pemetrexed	Phase II (recruiting)	NCT05299125 (AMIGO-1)
		Lazertinib + amivantamab + carboplatin/pemetrexed	Phase III (recruiting)	NCT04988295 (MARIPOSA-2)
		Lazertinib + amivantamab + bevacizumab	Phase II (recruiting)	NCT05601973 (AMAZE-Lung)
		Osimertinib+ EMB-01	Phase I/II (not yet recruiting)	NCT05498389
	MET ADC	Osimertinib or erlotinib + telisotuzumab vedotin	Phase I (active)	NCT02099058
HER2 alterations	HER2 ADC	Trastuzumab deruxtecan	Phase II (active)	NCT03505710 (DESTINY-Lung01)
	HER2 mAb	Osimertinib + necitumumab + trastuzumab	Phase I/II (recruiting)	NCT04285671
HER3 alterations	HER3 ADC	Patritumab deruxtecan	Phase II (recruiting)	NCT04619004 (HERTHENA-Lung01)
		Patritumab deruxtecan	Phase III (recruiting)	NCT05338970 (HERTHENA-Lung02)
		Osimertinib + patritumab deruxtecan	Phase I (recruiting)	NCT04676477
	Bispecific Antibody (EGFR-HER3)	Osimertinib + izalontamab	Phase II/III (recruiting)	NCT05020769
	EGFR-HER3 ADC (Bispecific Antibody)	Osimertinib + BL-B01D1	Phase II (not yet recruiting)	NCT05880706
AXL alterations	AXL TKI	Osimertinib + bemcentinib	Phase I/II (completed)	NCT02424617
Alterations affecting downstream molecules	BRAF inhibitor	Dabrafenib + trametinib	Phase II (recruiting)	NCT04452877
	mTOR inhibitor	Osimertinib + sapanisertib	Phase I (recruiting)	NCT02503722
		Osimertinib + sapanisertib	Phase I (not yet recruiting)	NCT04479306
	JAK inhibitor	Osimertinib + itacitinib	Phase I/II (active)	NCT02917993
		Osimertinib + golidocitinib	Phase I/II (completed)	NCT03450330 (JACKPOT1)
	MEK inhibitor	Osimertinib + selumetinib	Phase I (active)	NCT02143466 (TATTON)
		Osimertinib + selumetinib	Phase II (active)	NCT03392246
	PI3K inhibitor	Osimertinib + TQ-B3525	Phase I/II (recruiting)	NCT05284994
RET alterations	RET TKI	Osimertinib + selpercatinib	Phase II (recruiting)	NCT03944772 (ORCHARD)
ALK alterations	ALK TKI	Osimertinib + alectinib	Phase II (recruiting)	NCT03944772 (ORCHARD)
CDK4/6 amplification	CDK 4/6 inhibitor	Osimertinib + G1738 (lerociclib)	Phase I/II (completed)	NCT03455829
		Osimertinib + abemaciclib	Phase II (unknown)	NCT04545710
Others	Bcl-2 inhbitor	Osimertinib + navitoclax	Phase I (active)	NCT02520778
		Osimertinib + palcitoclax	Phase I (recruiting)	NCT04001777
	VEGF mAb	Osimertinib + bevacizumab	Phase II (active)	NCT03133546 (BOOSTER)
		Erlotinib + bevacizumab	Approved by the EMA in 2016	BELIEVE JO25567
		Osimertinib + bevacizumab	Phase I/II (completed)	NCT02803203
		Osimertinib + bevacizumab	Phase III (recruiting)	NCT04181060
		Osimertinib + bevacizumab	Phase III (recruiting)	NCT05104281
		Osimertinib + bevacizumab	Phase II (active)	NCT02971501
	VEGFR mAb	Osimertinib + ramucirumab	Phase II (recruiting)	NCT03909334
		Osimertinib + ramucirumab or necitumumab	Phase I (completed)	NCT02789345
		Erlotinib + ramucirumab	Approved by the FDA/EMA in 2020	NCT02411448 (RELAY)
	VEGFR-PDGFR-FGFR-cKIT TKI	Osimertinib + Anlotinib	Phase I/II (recruiting)	NCT04770688 (AUTOMAN)
	Aurora Kinase A inhibitor	Osimertinib + VIC-1911	Phase I (recruiting)	NCT05489731
		Osimertinib + alisertib	Phase I (not yet recruiting)	NCT04479306
		Osimertinib + LY3295668	Phase I/II (active)	NCT05017025
	MERTK and FLT3 TKI	Osimertinib + MRX-2843	Phase I (recruiting)	NCT04762199
	ROS17TRK/ALK TKI	Osimertinib + repotrectinib	Phase I (recruiting)	NCT04772235 (TOTEM)

The abbreviations used are: TKI, tyrosine kinase inhibitor; ADC, antibody-drug conjugate; mAb, monoclonal antibody; FDA, Food and Drug Administration of the United States of America; EMA, European Medicines Agency.

**Figure 3 cancers-15-05009-f003:**
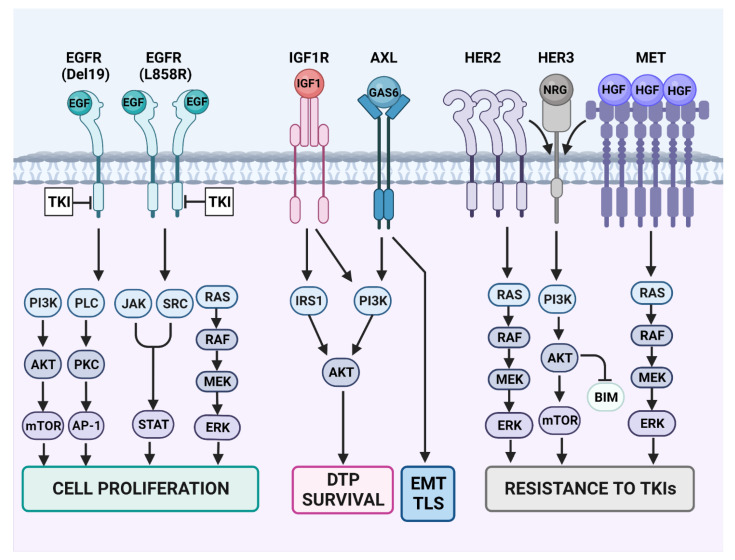
Resistance-conferring rewiring of RTK signaling in TKI-treated lung cancer. Some of the canonical signaling pathways engaged by EGFR are presented on the left side of the scheme. Note that EGFR dimerization is essential for EGF-induced activation of the wild-type kinase domain. Likewise, EGFR dimerization is crucial for activation of the L858R mutant form of EGFR’s kinase domain. However, all other mutant forms, including the different exon-19 deletion mutants (Del19), need no dimer formation for kinase activation. Because EGFR-specific TKIs can inhibit all EGFR downstream pathways, they usually confer sensitivity to the drugs. However, concomitant with blocking cell proliferation and inducing cell death, the TKIs prompt emergence of drug-tolerant persister (DTP) cells. Survival of the DTP cells is mediated by the insulin-like growth factor 1 (IGF1) pathway and the downstream insulin receptor substrate (IRS). Similarly, adaptive mutagenesis can be promoted by the GAS6-AXL pathway, which stimulates the SOS system (including inhibition of DNA repair and enhancement of trans-lesion synthesis (TLS) of DNA by error prone polymerases). AXL is also involved in another epigenetic program, the epithelial–mesenchymal transition (EMT). TLS and inhibition of DNA repair contribute to drug resistance by means of de novo mutagenesis of EGFR or downstream effectors like RAS and PI3K. Yet another therapy escape route is presented in the right part of the scheme. This route indirectly engages HER3, a kinase-dead member of the EGFR family that strongly activates the PI3K-AKT survival pathway. Constitutive activation of MET due to either *MET* gene amplification or overexpression of MET’s ligand, the hepatocyte growth factor (HGF), might occur in small populations of cancer cells, prior to treatment with TKIs. Similarly, constitutive activation of *HER2* due to either gene amplification, protein overexpression or the *HER2* exon 20 mutation may serve as a resistance mechanism, probably due to the tendency of this receptor to form heterodimers with HER3.

### 13.2. HER2 and HER3

EGFR family members, primarily HER2 and HER3, have repeatedly been implicated in resistance to EGFR inhibitors. HER2 amplification is found in higher percentages in resistant tumors treated with the first-generation TKIs, compared to those treated with osimertinib [42,91,93,94]. We previously demonstrated that, when blocking EGFR, not only HER2 but also its kin, HER3, underwent up-regulation that resulted in hyperactivation of the ERK pathway [143]. Accordingly, in animal models, triple combinations targeting EGFR (using both a kinase inhibitor and a monoclonal antibody), as well as HER2 (using a monoclonal antibody), showed efficacy in first- and in second-line treatment scenarios [144,145]. These observations suggested that the combination approach may be a suitable way to overcome resistance to EGFR inhibitors. In line with these findings, co-targeting EGFR and HER2 using antibody-drug conjugates (ADCs) might represent an alternative strategy. Thus, the combination of osimertinib and the HER2-specific ADC trastuzumab emtansine showed efficacy in overcoming resistance to osimertinib in EGFR-mutated lung cancer [146].

In similarity to HER2, HER3 expression is detected in a large fraction of NSCLC [147], and in EGFR-mutated lung cancers the levels are higher when compared to EGFR-wild type lung cancers [148]. In addition, HER3 overexpression has been observed in EGFR-mutated lung cancer models treated with osimertinib [149,150]. Along this line, it has been reported that resistance to EGFR TKIs can arise from HER3 via heterodimerization with other RTKs, such as HER2 and MET [151]. In this context, the anti-HER3 antibody patritumab is able to overcome TKI resistance mediated by neuregulin, the HER3 ligand, in EGFR-mutated lung cancer models [152]. Moreover, the levels of soluble neuregulin in the serum of patients with NSCLC appear to associate with better responses to patritumab [153]. In similarity to therapies targeting HER2 in NSCLC, ADCs directed against HER3 were also developed. Patritumab deruxtecan demonstrated both efficacy and safety in a phase I trial involving patients with EGFR-mutated lung cancer that progressed after TKI treatment [154]. Based on this encouraging result, the phase 3 trial, HERTHENA-Lung02, currently examines patritumab deruxtecan versus platinum-based chemotherapy in patients with EGFR-mutated NSCLC after failure of EGFR TKIs (NCT05338970).

### 13.3. Activation of AXL

AXL is overexpressed in several types of tumors, including lung cancer, and this feature is correlated with poor prognosis [155]. Additionally, AXL-positive tumors are more frequent among NSCLC tumors harboring mutant EGFR, compared to tumors expressing wild-type EGFR [156]. Overexpression of AXL has been found in EGFR-mutated lung cancer models that acquired resistance to the first-, second- or third-generation TKIs [45,110,124]. Of note, this receptor seems to be involved in the generation of DTPs, following treatment with EGFR TKIs [110,124]. Moreover, AXL is up-regulated in samples obtained from patients with EGFR-mutated NSCLC, after they developed resistance to TKIs [45]. In this respect, combination therapies involving EGFR TKIs and an inhibitor of AXL (either an antibody or a small molecule) were able to prevent resistance to EGFR TKIs in EGFR-mutated NSCLC models [123,157]. Consistent with these observations, several AXL inhibitors are being tested in clinical studies, in combination with EGFR TKIs [158].

### 13.4. IGF1-Receptor (IGF1R)

EGFR inhibition often leads to compensatory activation of the IGF1R pathway [159,160], with a potential benefit arising from the inhibition of this receptor [160]. In this regard, IGF1R is crucial for the establishment of DTPs following treatment of EGFR-mutated lung cancer cells with an EGFR-specific TKI [108,161]. Importantly, inhibition of IGF1R completely eliminates the ability of EGFR-mutated cancer cells to generate DTPs [108]. Thus, combinations of EGFR and IGF1R inhibitors might prevent the onset of resistance. Indeed, blocking both EGFR and IGF1R was able to inhibit EGFR-mutated lung xenografts [161]. Moreover, the insulin receptor substrate 1 (IRS1) has been shown to be crucial for the generation of DTPs, and combinations of EGFR and IRS1 inhibitors were highly effective in models of EGFR-mutated lung cancer that were examined in animals [162].

### 13.5. Fibroblast Growth Factor Receptors (FGFR)

Oncogenic fusions involving the fibroblast growth factor receptor (FGFR) have been reported to confer resistance to osimertinib [91,95]. In line with this, whole-genome CRISPR screening identified FGFR1 as the top target promoting survival of mesenchymal EGFR mutant cancers. Subsequently, testing combinations of EGFR and FGFR inhibitors revealed that the combination was able to overcome EMT-dependent resistance to EGFR-specific TKIs in models of EGFR-mutated lung cancer [163].

## 14. Conclusions

Although the two major oncogenes that drive NSCLC, *KRAS* and *EGFR*, activate largely overlapping biochemical pathways, their co-expression in tumors is very rare [164]. This mutual exclusivity is likely due to the toxic effects of their co-expression. In addition, the main carcinogen driving KRAS mutations is tobacco smoke, but the identity of the carcinogen(s) responsible for EGFR mutations is less characterized. Hence, it is predictable that prevention efforts will eventually decrease the absolute incidence of KRAS mutations, thereby increase the relative fraction of patients with EGFR-mutated NSCLC. Moreover, because KRAS^+^ tumors, better than EGFR^+^ tumors, respond to immune checkpoint inhibitors [165], the issue of resistance to EGFR inhibitors will likely remain one of the major challenges of medical oncology. Despite this gloomy scenario, the approval of osimertinib, the pioneer third-generation inhibitor, and its later application in first-line settings, promise that improved chemical designs, along with better understanding of the complex cascade that precedes emergence of new mutations, will enhance the success of future attempts to inhibit resistance to the TKIs. Unlike KRAS, which resides in the cytoplasm, the transmembrane localization of EGFR might become a key for effective targeting of the mutant forms using synthetic degraders, especially lysosome-targeting PROTACs [166]. In the same vein, the many lines of evidence linking up-regulation of compensatory RTKs (e.g., HER3 and HER2) and resistance to TKIs offer combinations of antibodies, for example cetuximab (anti-EGFR) plus trastuzumab (anti-HER2), as potential resistance-nullifying approaches. Likewise, the clinical approval of amivantamab, a bispecific antibody co-targeting MET and EGFR [138], is yet another source of hope that overcoming resistance to EGFR-specific TKIs is within reach.

## Figures and Tables

**Figure 1 cancers-15-05009-f001:**
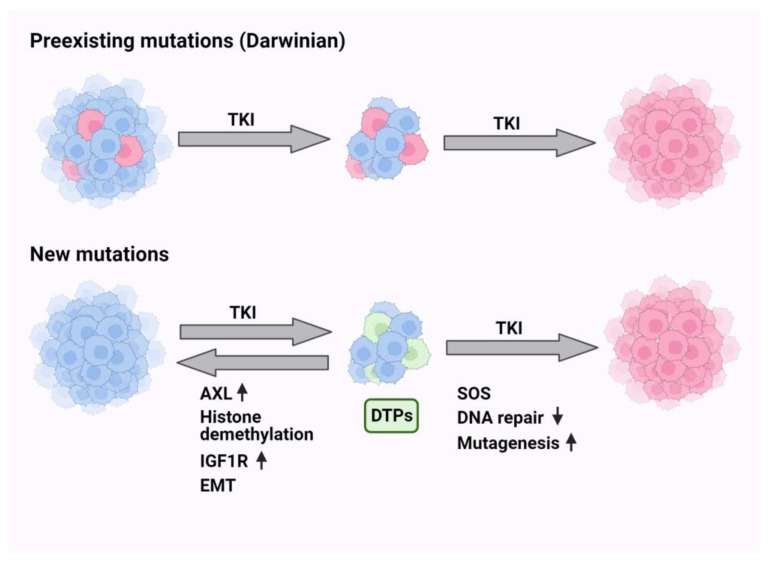
Darwinian and non-Darwinian models of resistance to tyrosine kinase inhibitors (TKIs). According to the Darwinian model (upper part), a few TKI-resistant cells preexist in tumors. Hence, treatment with TKIs kills the sensitive cells and repopulates the whole tumor with drug-resistant cells (shown in red). According to the alternative model, exposure to the drug induces the generation of reactive oxygen species (ROS) and extensive cell death. Concurrently, the drug instigates epigenetic changes, especially epithelial-mesenchymal transition (EMT), in a small population of cells called drug tolerant persister cells (DTPs). Note that this transition is reversible and involves histone demethylation, up-regulation of vimentin, AXL and additional RTKs. Continuous exposure to the drug elevates antioxidants, along with the SOS-like system. This system comprises down-regulation of both DNA repair and high-fidelity DNA polymerases, which are replaced by a group of low-fidelity (error prone) polymerases. Thus, DTPs might irreversibly acquire resistance due to on-target or off-target new mutations.

**Figure 2 cancers-15-05009-f002:**
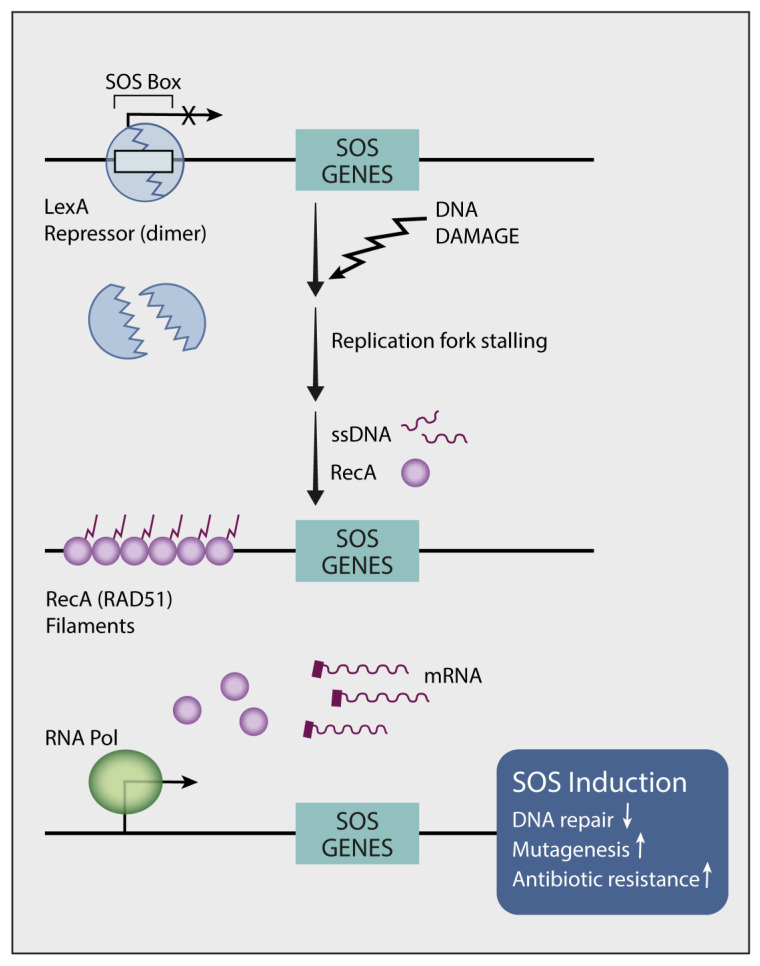
The bacterial SOS system. The outlined error-prone DNA repair system contributes significantly to DNA changes in bacteria. Normally, the dimeric LexA repressor binds with the SOS boxes of 20 nucleotide pairs found in the operator region of SOS genes, thereby preventing their action. Activation of the SOS genes occurs after genotoxic stress and DNA damage. Upon damage, newly formed single-stranded DNA (ssDNA) fragments that accumulate at stalled replication forks bind with RecA (the orthologue of RAD51 in mammals), an inducer. RecA forms a filament around these ssDNA regions. The ssDNA/RecA complex promotes auto-cleavage of the LexA repressor to permit recruitment of RNA polymerase and facilitate expression of genes under LexA control. Collectively, the newly translated proteins inhibit DNA repair while increasing mutagenesis, which confers resistance to antibiotics.

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
