# Peer review of "Resistance of Lung Cancer to EGFR-Specific Kinase Inhibitors: Activation of Bypass Pathways and Endogenous Mutators"

_cancers, 2023, doi:10.3390/cancers15205009_

Round 1
Reviewer 1 Report
The authors reviewed the genetic mechanisms behing the acquisition of drug resistance in lung cancer, as well as the use of TKI drugs for the treatment of such tumors. The review is well written and touches upon the most important subjects. It would be nice however, to add a couple of lines regarding the more common or severe adverse effects for such treatments.
Author Response
Reviewer #1
The authors reviewed the genetic mechanisms behing the acquisition of drug resistance in lung cancer, as well as the use of TKI drugs for the treatment of such tumors. The review is well written and touches upon the most important subjects. It would be nice however, to add a couple of lines regarding the more common or severe adverse effects for such treatments.
As requested, we added the following sentence in original page 4:
In comparison to chemotherapy, EGFR TKIs are relatively safe drugs. Nevertheless, several side effects have been reported. They frequently include skin effects and gastrointestinal tract toxicity, such as skin rash and diarrhea, respectively. More severe adverse effects include intestinal obstruction, hepatotoxicity and interstitial lung disease [1].
Reviewer 2 Report
Reviewer’s Comments:
This is a well-written, well-organized review article by Marrocco I and Yarden Y. However, it has certain caveats that need to be sealed before it can be published. Below are the comments yet to be addressed,
1. Table 1: The info about the drug(s) acceptability, whether it’s still under study, clinical trial, or revoked by certain other countries, needs to be mentioned at least in the footnote.
2. For the general readers, please explain sections 8 and 9 on page 7 ‘Darwinian’ and ‘Non-Darwinian’ mechanisms in a sentence or two at the beginning.
3. Authors must dedicate an entire section emphasizing the futuristic PROTAC technology, its unique feature, design, and mechanism of action highlighting recent studies targeting EGFR mutant NSCLCs or others, possibly before the conclusions section.
4. What is unique in this review, and why is it better than the numerous other reviews published based on a similar topic?
Needs minor editing
Author Response
This is a well-written, well-organized review article by Marrocco I and Yarden Y. However, it has certain caveats that need to be sealed before it can be published. Below are the comments yet to be addressed,
- Table 1: The info about the drug(s) acceptability, whether it’s still under study, clinical trial, or revoked by certain other countries, needs to be mentioned at least in the footnote.
Following this comment, we added the following information (in Table 1):
- Rociletinib, a 3rd generation TKI (covalent/irreversible), Del19/L858R/T790M-EGFR. Rejected by the FDA in 2016, NCT01526928, TIGER-1 (NCT02186301), TIGER-3 (NCT02322281) [2].
- Avitinib, also known as abivertinib.
- Olmutinib, a 3rd generation TKI (covalent/irreversible), Del19/L858R/T790M-EGFR, Approved in South Korea in 2016, NCT01588145, NCT02485652 [3]. Terminated because of two cases of toxic epidermal necrolysis, one of them fatal.
- Naquotinib, a 3rd generation TKI (covalent/irreversible), Del19/L858R/T790M-EGFR, Clinical Phase III (terminated), NCT02588261 [4].
- For the general readers, please explain sections 8 and 9 on page 7 ‘Darwinian’ and ‘Non-Darwinian’ mechanisms in a sentence or two at the beginning.
The following sentences were added at the beginning of section 8:
In the context of anti-cancer therapies, Darwinian mechanisms of drug resistance refer to a selection process that relies on pre-existing genetic variation that is present in a cancer cell population before starting the treatment. The pressure applied by the drug allows cells expressing a resistance-conferring mutation to survive the treatment (selection), leading over time to the establishment of a drug-resistant cancer cell population. Darwinian selection was observed, for instance, in patients with KRAS wild-type colorectal cancer that developed resistance after treatment with the anti-EGFR monoclonal antibody panitumumab [5]. While KRAS mutations were not found prior to the treatment, they became detectable 5-6 months after treatment with the antibody.
The following sentence was added at the beginning of section 9:
Non-Darwinian selection includes all mechanisms that do not involve genomic changes. In this context, epigenetic alterations, factors from the microenvironments, cellular plasticity and phenotypic adaptation are phenomena that might lead to drug resistance.
- Authors must dedicate an entire section emphasizing the futuristic PROTAC technology, its unique feature, design, and mechanism of action highlighting recent studies targeting EGFR mutant NSCLCs or others, possibly before the conclusions section.
As requested, the following paragraph was added after the paragraph dealing with 4th generation EGFR TKIs:
PROTACs
In parallel to the development of 4th generation EGFR TKIs, a major current effort is directed towards clinical development of proteolysis targeting chimeras (PROTACs), representing an alternative strategy to directly target EGFR. PROTACs consist of two covalently linked protein ligands: one binds with the target protein (the protein intended to be degraded), while the other molecular arm recruits an E3 ligase that promotes ubiquitination and degradation of the target protein [6]. Notably, the numbers of PROTACs that are currently being evaluated in vitro, in vivo and in clinical studies is increasing [7]. Several PROTACs targeting EGFR have been developed (reviewed in [8]). By using EGFR TKIs as the target protein binder, it is possible to selectively target mutated forms of the receptor, while sparing the wild type form of EGFR. In this regard, two PROTACs synthetized by using gefitinib as the EGFR binder were able to reduce proliferation of Del19- and L858R-EGFR expressing NSCLC cells, along with the levels of EGFR and the activation of downstream signaling pathways [9]. In a more recent study, Vartak et al. developed a PROTAC containing cetuximab as the EGFR binder, thereby reducing viability of EGFR-mutated NSCLC cells [10]. Despite the promising results showing in vivo efficacy of EGFR-specific PROTACs [11], additional preclinical data are needed prior to translating these findings to clinical applications. For example, membrane permeability and metabolic stability are crucial factors that need to be considered before attempting clinical development of EGFR-specific PROTACs.
- What is unique in this review, and why is it better than the numerous other reviews published based on a similar topic?
The review we offer contains an updated summary of the most recent studies dealing with EGFR TKIs, with a special focus on mechanisms of drug resistance. In this regard, our literature review brings together papers dealing with adaptive mechanisms involving RTK-mediated resistance to EGFR-specific TKIs. At the same time, we shed light on the roles played by drug-tolerant persister cells (DTPs), as well as on the role of intrinsic mutators in the onset of resistance to EGFR TKIs. Moreover, our review highlights the recent discovery of L858R-EGFR as a potential biomarker predicting response to monoclonal antibodies directed against EGFR (e.g., cetuximab). In addition, the two tables we incorporated provide an updated list of all EGFR TKIs that were either approved or those that are still under preclinical/clinical evaluation (Table 1). Also listed are mechanisms of resistance to EGFR TKIs and the drug combinations that are currently being clinically evaluated with the goal of overcoming specific mechanisms of resistance (Table 2). Lastly, we cite a recent study published in Nature by Charles Swanton and colleagues, who have shown that air pollutants promote lung cancer by acting on lung epithelial cells that pre-express EGFR mutations. The following text was added to the review (Page 3. line 115):
Another factor that might increase incidence of EGFR-mutated lung cancer appears to be air pollution. Hill et al. showed that exposure to particulate matter measuring ≤2.5 μm (PM2.5), promotes lung cancer development by recruiting macrophages into the lungs, which in turn release IL-1β. Consequently, a progenitor-like cell state is induced in lung alveolar type II epithelial cells harboring pre-existing EGFR activating mutations [12].
Reviewer 3 Report
Parts 2-7.2 needs more thorough understanding by the authors of the structure and ideas that authors want to deliver. Starting from chapter 4-7.2 the text seems a list of poorly linked facts rather than a structured statement. In our opinion the review is not benefiting from such a long introduction to the problem.
2 48-55 paragraph is not very clear and the transition from one alteration type to another does not seem logical.
2 57-67 the idea of this part is not clear. While amphiregulin level for example might be correlated to EGFR MAbs efficacy it is not used as a biomarker.
3 107-109 - the fact about BRAF TMB and IO sensitivity should be verified
Part 8-9
In these parts authors at last reach the question of resistance to TKI and discuss an interesting question whether acquired resistance is a selection of clones or appearance of new subpopulations. While the ideas founded here are essential for the whole review, authors cite a number of significant studies elucidating various features of persister population, this part might benefit from more clear conclusion and language.
Part 10 seems very interesting introducing not so widely known data on mutators and mechanisms of adaptation in DTP cells. Still one can hardly link this idea to previously expressed points. Probably this chapters needs restructuring and emphasis on the link with previous parts.
Part 12 includes interesting and important points on selection of the treatment approach based on predictive markers, that is still not used for TKI naive EGFR mutation careers. This part seems pulled out of context in the end and should be moved to the begging.
In general, review needs to be restructured with shortened introduction and part concerning primary treatment.
minor corrections
Author Response
Parts 2-7.2 needs more thorough understanding by the authors of the structure and ideas that authors want to deliver. Starting from chapter 4-7.2 the text seems a list of poorly linked facts rather than a structured statement. In our opinion the review is not benefiting from such a long introduction to the problem.
In response to this critical comment, we trimmed the text of Part 2 through Part 7.2. For example, in Part 2, we deleted the following text:
The key for this strategy is the development of proteolysis targeting chimeras (PROTACs), consisting of two arms, one binds with the protein of interest and the other recruits an E3 ubiquitin ligase. This configuration enables PROTACs to recruit the E3 ligase to specific targets, such as RTKs, thereby leading to proteasomal degradation of the receptor of interest, for example EGFR [13]. Especially interesting are PROTACs targeting specific mutant RTK proteins, such as a triple mutant form of EGFR [14].
In addition, in Part 7 we deleted the reference to NRAS/KRAS and shortened the whole part, such that the revised version includes no breakdown to 7.1 and 7.2.
2 48-55 paragraph is not very clear and the transition from one alteration type to another does not seem logical.
To improve clarity and transitions, we rewrote the corresponding paragraph, which reads as follows in the revised manuscript:
The most frequent gene aberration of HER2/ERBB2, the closest homologue of EGFR, is gene amplification, which occurs in breast, gatsric and other tumors, but rare missense mutations have also been reported [15,16], Importantly, very high expression levels of HER2 predict response to HER2-targeting monoclonal antibodies (mAbs), such as trastuzumab/Herceptin. Similarly, overexpression of EGFR occurs in approximately 50% of brain tumors of glial origin due to gene amplification [17], and a large fraction of these tumors also present internal deletions within the extracellular domain [18]. However, these observations have not been translated to EGFR targeted treatments of brain malignancies.
2 57-67 the idea of this part is not clear. While amphiregulin level for example might be correlated to EGFR MAbs efficacy it is not used as a biomarker.
As requested, we clarified this point in the revised text. The corresponding new text is shown below:
The other oncogenic roles of growth factors and RTKs involve autocrine or paracrine loops that engage a growth factor and a specific RTK. For example, high expression of EGFR ligands, especially amphiregulin, supports proliferation of CRC cells and might predict response to a combination of chemotherapy and an anti-EGFR mAb, such as cetuximab [19]. However, the abundance of amphiregulin in CRC specimens has not been translated to a predictive biomarker of response to anti-EGFR antibodies.
3 107-109 - the fact about BRAF TMB and IO sensitivity should be verified.
In response to this comment we revised the corresponding text and added a new reference that supports the point dealing with TMB and IO sensitivity. The revised text reads as follows:
Although immunotherapy combinations, for example the combination of antibodies targeting CTLA-4 and PD1, seem to benefit some patients presenting with advanced or metastatic NSCLC, tumors with EGFR mutations in general lack abundant infiltrating lymphocytes and have a relatively low tumor mutational burden [20]. Hence these tumors tend to not respond to immune checkpoint blockers. This contrasts with most KRAS- and BRAF-mutated NSCLCs, which are associated with a higher mutational burden [21].
Part 8-9
In these parts authors at last reach the question of resistance to TKI and discuss an interesting question whether acquired resistance is a selection of clones or appearance of new subpopulations. While the ideas founded here are essential for the whole review, authors cite a number of significant studies elucidating various features of persister population, this part might benefit from more clear conclusion and language.
In response to this comment, we improved the language and added clearer conclusions at the ends of both Part 8 and Part 9. The conclusion that appears in the end of Part 8 reads as follows:
In conclusion, the examples from lung cancer and leukemia exemplify the role played by pre-existing mutations and intra-tumor heterogeneity in resistance to anti-cancer drugs, such as kinase inhibitors.
The conclusions that appear in the end of Part 9 read as follows:
In summary, the non-Darwinian mechanisms of drug resistance and especially the emerging common functional features of DTP cells, exemplify the robust and diversified nature of drug tolerance. This offers a spectrum of molecular targets, such as notch and AXL, for combination treatments able to nullify or significantly delay the onset of resistance to drugs, including EGFR inhibitors.
Part 10 seems very interesting introducing not so widely known data on mutators and mechanisms of adaptation in DTP cells. Still one can hardly link this idea to previously expressed points. Probably this chapters needs restructuring and emphasis on the link with previous parts.
To link Part 10 and the previous parts (Parts 8 and 9), we added the following text at the beginning of Part 10:
- Endogenous mutators promote the emergence of new mutations while under TKI treatment
In vitro studies uncovered two routes of resistance to TKIs: (i) a Darwinian route that selects specific clones, which are drug resistant due to a pre-existing secondary mutation, and (ii) a less understood route that promotes de novo emergence of resistance-conferring mutations [22]. Notably, the mechanisms underlying emergence of the secondary (e.g., T790M) and tertiary (e.g., C797S) EGFR mutations are distinct from the mode that generates primary EGFR mutations. The primary mutations likely pre-exist in normal lung tissues but they are later exposed due to the action of tumor promoters, such as air pollutants [23]. What mechanisms could underlie the apparent drug-induced accelerated evolution? In 1975 Miroslav Radman reported an inducible bacterial DNA mutagenesis system, the SOS response (see Figure 2), which might explain the link between genotoxic stress and adaptive mutagenesis [24].
Part 12 includes interesting and important points on selection of the treatment approach based on predictive markers, that is still not used for TKI naive EGFR mutation careers. This part seems pulled out of context in the end and should be moved to the begging.
As requested, we moved up the part dealing with the potential predictive biomarker (L858R). The new order of parts is shown below:
- Endogenous mutators promote the emergence of new mutations while under TKI treatment
- A biomarker predicting response of EGFR+ tumors to antibody rather than TKI treatment
- Individual bypass mechanisms and the respective combination therapies (Figure 3 and Table 2)
13.1. MET activation
13.2. HER2 and HER3
13.3. Activation of AXL
13.4. IGF1-receptor (IGF1R)
13.5. Fibroblast growth factor receptors (FGFR)
- Conclusions
In general, review needs to be restructured with shortened introduction and part concerning primary treatment.
Round 2
Reviewer 2 Report
All the questions raised by this reviewer have been addressed.
Need a minor language check before possible acceptance.
Reviewer 3 Report
Thank you for addressing the issues.